# HUSH (Hiking in Urban Scientific Heritage): The Augmented Reality for Enhancing the Geological and Naturalistic Heritage in Urban Areas

Laura Melelli *, Giulio Bianchini and Livio Fanò

Department of Physics and Geology, University of Perugia, 06123 Perugia, Italy;
giulio.bianchini@studenti.unipg.it (G.B.); livio.fano@unipg.it (L.F.)
* Correspondence: laura.melelli@unipg.it; Tel.: +39-075-5849579

**Featured Application: This work describes HUSH, a mobile application designed for the promotion of geoheritage, focusing specifically on the naturalistic values within urban areas.**

**Abstract:** Over the past two decades, significant efforts have been made to diversify the tourism industry and enhance its educational experience. One such endeavor is urban trekking and geotourism, which have emerged as important means of promoting geological knowledge. The recent advancements in augmented reality technologies as well as the increasing availability of 'born digital' data such as those gathered from social media create a basis for the development of immersive and customized touristic experiences. Urban scientific heritage, augmented reality, and data mining are the key elements of the HUSH project. Its first focus is the identification of the naturalistic components in a given urban area (flora, fauna, and geological features) through literature surveys and scientific research. These factors become points of interest (PoIs) along touristic paths, where they are connected to the historical and artistic components of the area. Augmented reality serves as the medium through which the user can access this content. The contents are delivered as videos, text, images, or interactive 3D models. The mobile application from this project is a useful tool for promoting geoheritage and naturalistic values in urban areas and for improving the awareness and the sustainability of our cities.

**Keywords:** geoheritage; augmented reality; mobile application; geotourism; natural heritage; urban geology

## 1. Introduction

More than half of the world's population currently resides in urban areas, and according to the United Nations, this number is projected to increase to two out of three people by 2050 [1]. Urbanization is a process affected by several constraints and issues concerning sustainability, emphasizing the need for nature-based solutions to prevail [2]. Prioritizing green solutions is crucial to increase the quality of life in cities through the improvement of sustainability. The presence and the awareness of abiotic and biotic parameters in urban and suburban areas is, in a broader vision, one of the most viable green solutions and serves as the initial step toward progress [3,4]. The natural elements present in a city, which are declining in different aspects, are the geological characteristics of the settlement and the flora and fauna inhabiting green areas within cities [5–7].

The Investigation of the abiotic components of cities starts by considering that the establishment of an urban area is always influenced by geological factors [8–10]. Site selection for urban development derives from a well-defined set of variables. Among them, especially in ancient settlements, geological factors such as the topographic arrangement, the bedrock composition, and the water availability are all determining factors. The

location is the primary parameter taken into account, as a favorable and strategic location can ensure efficient communication with other cities and increase trade and cultural exchanges. To achieve both of these conditions, morphological conditions (plain areas easily practicable) and hydrographic conditions (proximity to navigable rivers) play a key role. The topographic layout has a defensive function, minimizing also vulnerabilities to natural hazards such as landslides or floods. Underground resources such as lithotype outcroppings at a settlement site are a crucial asset for obtaining building stones and can provide water supplies where aquifers are present. All these preliminary conditions are shared in a common geological component. For this reason, it is both easy and meaningful to promote and disseminate geological content to foster a different perspective and enhance awareness of urban areas. The geological and geomorphological conditions not only provide opportunities but also establish constraints on urban development, imposing boundaries between man-made and natural environments. This is particularly true in downtown areas with recent suburban development, where new technologies have allowed people to overcome the barriers imposed by natural conditions. In the past, the geological background in urban areas has been investigated and interpreted in relation to natural hazards and derived risk conditions such as flooding, landslides, or volcanic and seismic events [11–17]. Furthermore, in cases where urban development is constrained by morphological limits, natural green areas are preserved; these zones can be a precious resource for biodiversity preservation. The fauna and flora that populate our cities are surprising and largely not known, but they represent important and underestimated content. Actually, there is increasing attention to natural sciences as a tourist and educational resource within urban areas. The exploration of geoheritage and the biotic component of cities has emerged as a promising and exciting field of investigation within the Earth sciences [9,10,18–20]. In addition, biotic presence is often undervalued because of the prevalent perception of the anthropic component and because historical and architectural values predominate in touristic themes, but this presence can be a meaningful component of the perception of urban spaces, if properly disclosed [21].

This article presents a case study focused on the promotion of naturalistic heritage within the urban area of Perugia, the capital town of the Umbria region (Central Italy). In the Umbria region, despite great geodiversity characterizing the territory [22], the most important towns have a common geological setting and are an excellent example for promoting the geotouristic approach. In the region, mountains and hilly areas prevail, and only a small percentage of the territory is occupied by plain areas, corresponding to the main fluvial valleys and to intermontane basins. With this morphological arrangement, the most relevant historical cities are placed on the top of gentle hills, bordering the intermontane basins of the region [23]. Some of these locations have been selected as initial settlement since the X century B.C., dating back to the Etruscan and Roman periods and with meaningful developments in medieval times. The higher altitude values ensured better environmental conditions compared to the lowlands, where swamps and flooding, mostly in past centuries, could foster malarial conditions. At the same time, the proximity to the flat areas was fundamental for closeness to communication networks. These hilly areas are made by fluvial and lacustrine deposits (Pliocene–Pleistocene) sedimented by the drainage network flowing from the surrounding mountain areas to the main tributaries present in the valleys below. The fluvial deposits guaranteed a water supply, the possibility of agricultural use, and the opportunity to create underground cavities for different purposes such as storage and tunnels, but also wells and tanks, as well as tombs and necropolises [24,25]. However, the morphologic configuration was affected by natural hazards such as landslides, imposing limits ton urbanization. These areas, free from urban fabric, have become privileged niches for wildlife and spontaneous flora. They can therefore be optimal laboratories for the observation of naturalistic values in a city. For their location, the geological and geomorphological conditions, and the great geodiversity and biodiversity of the territory, the small towns in the Umbria region are an excellent example of promoting natural heritage. In addition, as is common in urban areas, the presence of efficient digital

infrastructure facilitates the dissemination strategy [26–29]. Mobile applications have a broad spectrum of uses in urban environments [30], from environmental modeling [31] to optimization of mobility [32]. In addition, mobile applications are a perfect tool for exploiting digital and multimedia materials for didactic and touristic purposes, engendering the possibility to create an immersive experience and expanding the temporal and spatial scales of perception that you can have simply observing a point of interest [29,33]. In several case studies mobile applications have already been created and tested with urban areas as prime locations [34] but also in non-urbanized contexts [35–39]. The topic is often focused on building stones, since the target images are easily recognizable and obtainable by visual recognition tools [40]. Some examples of areas where mobile applications have been tested in Italy include cities such as Rome [34], Turin [41,42], and Naples [43], as well as in the Molise region [44]. Other examples are present in different geographical areas too, such as in Poland [45] the Czech Republic [35], and China [46].

In this paper, Perugia is proposed as a test area for promoting the naturalistic heritage present, as the urban context is similar to other towns of Central Italy (Figure 1).

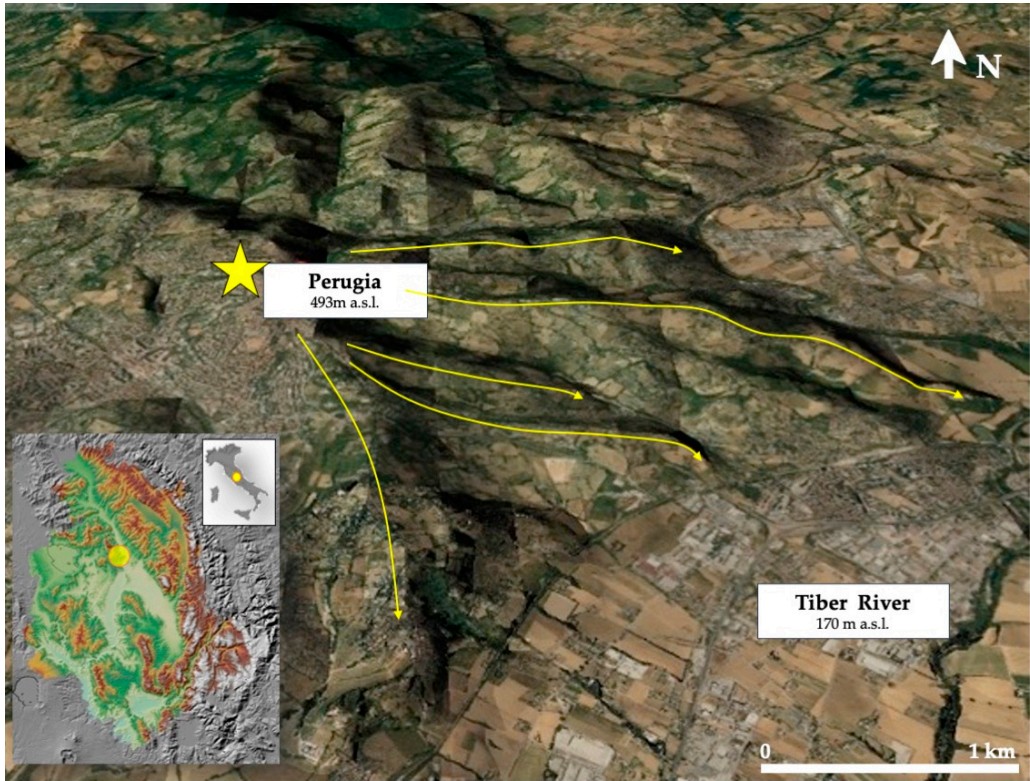

**Figure 1.** Location map. Perugia in the Umbria region and the hill of Perugia. The position of the downtown area is marked by the star, and the arrows highlight the direction of the ridges moving from the downtown area toward the Tiber River valley below.

A mobile application named HUSH (Hiking in Urban Scientific Heritage) is illustrated as a tool that exploits the potential of digital multimedia content and augmented reality, enhanced by machine learning techniques. This project aims to valorize naturalistic heritage, working better when such heritage is close to historical–artistic monuments in an urban area, and tries to involve citizens and tourists. Moreover, HUSH tries to support the local economy, making known the typical products and activities. The idea behind HUSH, the selection and creation of the content, and the infrastructure of the mobile application are described in the following paragraphs.

## 2. Methodology

The aim of a mobile application designed for the promotion of naturalistic heritage in urban areas is based on the assumption that even if the touristic and educational activities in cities are focused on historical–artistic values, a city may be a domain of naturalistic values too. Determining what these values are and where they can be found is the first step of the HUSH project. The description of the naturalistic values implemented in the application, the content, and the translation into digital information are described below.

The geological heritage has been classified into three main groups: the building stones, the geomorphological features, and the hydrogeological and hydrographic resources. Starting from these three groups of points of interest (hereinafter PoIs), it is possible to analyze the abiotic component using both the endogenous parameters and the exogenous modeling agents responsible for the urban development of the city. The faunistic and floristic heritage has been explored, taking into account the fauna and the flora present in the downtown area and in the suburban areas too.

### 2.1. Geological Points of Interest

#### 2.1.1. The Building Stones

In the Perugia downtown area, the facades of the most relevant historical buildings are made of stones that, with some exceptions, are derived from the settlement area or from the surrounding ones. For this reason, even if the urbanization covers the natural outcrops, the observation of the building and ornamental rocks reveals important and unexpected geological information on the site and surrounding areas [47,48].

Observing a building stone is a good exercise to recognize the mineralogical and petrographic characteristics of a rock. A single block, cut on differently oriented surfaces, can show several orientations of the content, such as minerals, fossils, and textures. In addition, starting from a single lithotype, it is possible to enlarge the information to the level of knowledge of the regional geology in a different spatial scale. Beginning from where a particular rock type outcrop exists, it is possible to understand the geological evolution of a wider area and the reason for the formation and the presence of the different lithotypes. Some stone blocks may present paleontological subjects too, adding information beyond the mineralogical and petrographic aspects.

#### 2.1.2. The Geomorphological Evolution

Looking for landforms in an urban area, the initial phase is to investigate the topographic layout. To achieve this, the first step is rigorous bibliographic research with a multidisciplinary approach. Historical manuscripts, ancient paintings, and maps are the first types of documents that should be considered, as the actual city is the result of complex transformations over the centuries [49,50]. Even if the urban context discourages the observation of the natural terrain, fieldwork can identify rare outcrops or confirm the presence/absence of man-made filling materials. Valuable support is derived from the anthropic underground cavities that in several Umbrian cities, and in Perugia too, are widespread in the downtowns. These places allow for the observation of natural terrains without vegetation or other types of impediments that are often present on the topographic surface. Moreover, having these cavities beneath the perimeter walls along at least two orthogonal directions, the characteristics of the lithotype can be observed from different points of view. This allows for better comprehension of the petrographic and sedimentological characteristics [24,25,51]. A last contribution derives from the interpretation and correlation of the geognostic surveys, generally numerous in anthropized contexts, providing information in areas where other types of data (surface or subterranean) are missing [49,52–54].

For the city of Perugia, ancient maps reproducing the evolution of the downtown area have been collected; the oldest is dated to 1572, while the newest one is dated to the XIX century. A detailed collection of field observations (geological and geomorphological) has been acquired. About 100 geognostic surveys were gathered, allowing the localization and

estimation of the thickness of anthropic deposits. By knowing the base of the filling deposits in a set of points, it is possible to reconstruct by interpolation the pre-settlement topographic surface. All these data have been georeferenced and compared in a GIS project, and with a geostatistical analysis, a digital evolution model was derived. This first DEM imitates the initial topographic layout before human settlement. Then, a second DEM, reproducing the actual surface and with the same horizontal resolution and spatial reference system, was derived. This second DEM is obtained by the interpolation of contour lines from a topographic map at a scale of 1:10,000. By subtracting the two DEMs, the landscape's evolution can be simulated. The resulting output grid highlights the filled areas (positive values), while the excavated zones correspond to the negative values. This information reveals the original and true morphological structure of the urban areas, which are usually perceived in a different way. Knowing about the transformations that occurred over the centuries can change the awareness of urban places, enriching the cultural background of visitors.

### 2.1.3. The Hydrogeological Evidence: Hydraulic Works

Groundwater and surface water resources are a necessary condition for the selection of the settlement site. In the city of Perugia, ancient wells, tanks, fountains, and drainage tunnels have been numerous since the Etruscan age (at least since the VIII century B.C.). Some of the most valuable works have been selected and inserted as points of interest. On the surface, the drainage network is a useful parameter for the palaeogeographical reconstruction of the settlement area. In Perugia city, the morphological arrangement can be simplified into ridges and valleys, with the valleys created and occupied by streams flowing from the top of the downtown area toward the bottom of the relief. In addition, the slopes facing eastward and northward have higher slope angle values, while the western and southern ones are gentle, dipping toward the fluvial valleys below. This different morphological set-up has affected the development of the city, with the areas toward the west being the only ones available. For this reason, the rivers flowing westward have been covered over the last few decades. On the contrary, the streams flowing eastwards, short and steep, show longitudinal profiles far from equilibrium and with strong headwater erosion, with drainage basins populated with wildlife and spontaneous vegetation. Headwater erosion is a trigger for river erosion and mass movements, and for this reason these streams are a good topic for some PoIs. The western streams are not visible anymore and are an intriguing theme for palaeogeographical reconstruction of the city area. The drainage network may provide a starting point for topics such as hydrology, applied geology, and issues related to natural hazard and flooding.

### 2.2. Naturalistic Point of Interest

In the urban areas the biotic component, such as wildlife or spontaneous vegetation, is present but not always very visible. Regarding the floristic heritage, native vegetation is present along with an increasing presence of alien vegetation. In some cases, such as the monumental trees, the presence of a floristic heritage is already known, but, expanding our point of view towards the plant world, the observation of urban green areas is an effective tool for didactic and touristic purposes. In the city of Perugia, 106 trees have been labeled as monumental, and 22 green areas are present, classified as urban parks and urban vegetable gardens. Moreover, the drainage basins of the rivers flowing east and north, as already stated, are in a natural state, hosting spontaneous vegetation.

Similarly, even animals can be classified into two groups: a first set that has perfectly integrated with the urban fabric, such as squirrels; and a second group of wild animals (such as foxes, for example) living in the peri-urban areas. In particular for the fauna, which unlike plants are in motion and rarely observable in live view, the use of multimedia tools is an optimal choice for knowledge regarding them and its dissemination. A set of wildlife cameras with infrared sensors has been positioned both along the rivers' catchments and in the downtown area, in urban gardens close to natural areas.

The simple observation of a place that has changed over the centuries, of a rock on the wall of a historic building, an ancient well, or a tree along a city street, cannot provide scientific information in a simple and straightforward manner. Scientific subjects are certainly more difficult to convey to the general public. For this reason, it is necessary to adopt innovative and more attractive strategies to engage an increasingly large audience.

### 2.3. The Building of PoIs (Points of Interest)

Each feature worthy of interest has been classified as a point of interest (PoI). For each PoI, survey and sampling, scientific investigation, and digital content creation have been undertaken. PoIs are divided into three big macro-categories: geological, biotic (flora and fauna), and commercial, with the third one having the aim of promoting the local economy and providing support for urban trekking. The macro-categories have a numeric or logical value depending on the type of content. In the database, sub-categories are preset too, i.e., the class "geological" has the sub-categories petrographic, geomorphologic, and hydrogeologic. Another example of a sub-category is the class "type of place", which can be artistic, spiritual, etc. (see Figure 2 as example).

**Table 1.** Classification of the Points of Interest.

| N. | Name | X Loc | Y Loc | Category | Sub-Category | Path |
|---|---|---|---|---|---|---|
| 1 | Biscarini Furnace and Laboratory | 43°06′24″ N | 12°23′30″ E | Geological | Petrography | 2 |
| 2 | Bomb shelter S. Ercolano | 43°06′31.13″ N | 12°23′22.38″ E | Geological | Sedimentology | 2 |
| 3 | Botanic Garden | 43°05′49.13″ N | 12°23′48.26″ E | Naturalistic | Flora (threes) | 2 |
| 4 | Bulagaio Stream | 43°06′52″ N | 12°23′22.8″ E | Geological | Geomorphology | 1 |
| 5 | Cavour Street | 43°06′31.13″ N | 12°23′22.38″ E | Naturalistic | Fauna (squirrel) | 2 |
| 6 | Etruscan Arch | 43°06′52″ N | 12°23′22.8″ E | Geological | Petrography | 1 |
| 7 | Fountains | 43°06′31.13″ N | 12°23′22.38″ E | Geological | Hydrogeology | 2 |
| 8 | Garibaldi Street | 43°07′10.4″ N | 12°23′04.6″ E | Naturalistic | Fauna (gecko) | 1 |
| 9 | Giordano Bruno Square Well | 43°06′24″ N | 12°23′30″ E | Geological | Hydrogeology | 2 |
| 10 | Grimana Square | 43°06′52″ N | 12°23′22.8″ E | Geological | Geomorphology | 1 |
| 11 | Medieval Garden | 43°06′05″ N | 12°23′44″ E | Naturalistic | Flora (threes) | 2 |
| 12 | Orto Sole | 43°06′52″ N | 12°23′22.8″ E | Naturalistic | Fauna (bee) | 1 |
| 13 | S. Agostino Church | 43°06′58.2″ N | 12°23′24.53″ E | Geological | Petrography, Paleontology | 1 |
| 14 | S. Angelo Gate | 43°07′10.4″ N | 12°23′04.6″ E | General | | 1 |
| 15 | S. Angelo Church | 43°07′10.4″ N | 12°23′04.6″ E | Geological | Petrography, Paleontology | 1 |
| 16 | S. Domenico Church | 43°06′24″ N | 12°23′30″ E | Geological | Petrography | 2 |
| 17 | S. Domenico Church | 43°06′24″ N | 12°23′30″ E | Naturalistic | Fauna (bat) | 2 |
| 18 | S. Margherita Stream | 43°06′16.6″ N | 12°23′34.4″ E | Geological | Geomorphology | 2 |
| 19 | S. Margherita Stream | 43°06′16.6″ N | 12°23′34.4″ E | Naturalistic | Fauna | 2 |
| 20 | Solfaroli Street | 43°07′10.4″ N | 12°23′04.6″ E | Geological | Minerology | 1 |
| 21 | S. Pietro Gate | 43°06′16.6″ N | 12°23′34.4″ E | Geological | Petrography | 2 |
| 22 | S. Pietro Gate | 43°06′16.6″ N | 12°23′34.4″ E | General | | 2 |

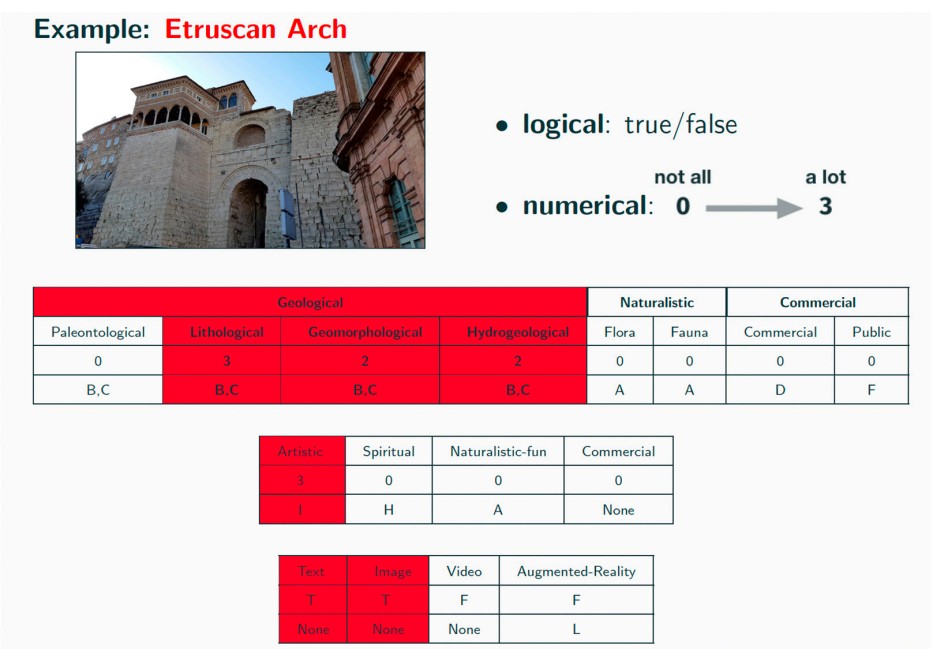

**Figure 2.** The categorization of a PoI (the Etruscan Arch, point 6 in Tables 1 and 2). The capital letters assign a category to each PoI derived from the user's profile and intended for the algorithm. A: naturalistic, B: geographic, C: scientific, D, E: shops, F, G: public exercise, H: religious, I: artistic, historical, cultural, L: playful, computer based. T: true, F: false.

**Table 2.** Tools used for each PoI.

| N. | Photo | Map | Video/GIF | Drone Flight | 3D Model | Virtual Flight | Text |
|---|---|---|---|---|---|---|---|
| 1 | √ | √ | | | | | √ |
| 2 | √ | √ | | | √ | | √ |
| 3 | √ | √ | | | | | √ |
| 4 | √ | √ | √ | √ | | | √ |
| 5 | √ | √ | | | √ | | √ |
| 6 | √ | √ | √ | | | | √ |
| 7 | √ | √ | | | | | √ |
| 8 | √ | √ | | | | | √ |
| 9 | √ | √ | | | | | √ |
| 10 | √ | √ | √ | | | | √ |
| 11 | √ | √ | √ | | | | √ |
| 12 | √ | √ | √ | | √ | | √ |
| 13 | √ | √ | | | | | √ |
| 14 | √ | √ | | | | √ | √ |
| 15 | √ | √ | | | √ | | √ |
| 16 | √ | √ | √ | | | | √ |
| 17 | √ | √ | √ | | √ | | √ |
| 18 | √ | √ | | √ | | | √ |
| 19 | √ | √ | √ | | | | √ |
| 20 | √ | √ | | | | | √ |
| 21 | √ | √ | | | | | √ |
| 22 | √ | √ | | | | √ | √ |

According to their specific characteristics, for PoIs' description and visualization, different multimedia tools have been selected such as images, photos with hyperlinks, or videos with subtitles. The videos should be descriptive of a specific argument and can be made up of clips of other videos already created, combined with images taken on site. Otherwise, in particular for large areas and for the PoIs with geomorphological value, drone footage has been acquired. Shooting from above has the advantage of showing wide features and effectively visualizes the surface topography and, where necessary, the proximity of the urban fabric and the green areas where naturalistic values can be more present. Thematic maps with hyperlinks have been added too. An interactive map is a more appealing document. It is well-known that maps are not always easy to read and to understand for non-experts. The possibility to query the map, choosing only some aspects, should encourage users' fruition. Where the value expressed in the PoI is an object with a well-defined geometry, such as a fossil or a mineral, a 3D model has been built. The virtual 3D models allow interaction because the user can manipulate them by virtually rotating or dividing the object into smaller parts, obtaining a more interesting conception and visualization.

The material has been included in webpages edited in WordPress, as it is an open-source content management system that is easy to use and to update. PoIs have been connected along two itineraries following some of the main historical streets of the Perugia city center. Along each of these paths, both geological and flora and fauna PoIs are present.

*2.4. The Creation of the Mobile Application*

The app creation can be summarized in three steps: the database creation of user data and PoI data; the back end with queries or scripts; and the front end that defines access to the scientific contents of the POIs for the users. The application has been developed with the IDE Unity3D, a complete and powerful tool for developing cross-platform applications.

The flow of the app is very simple and is divided into three main parts (Figure 3).

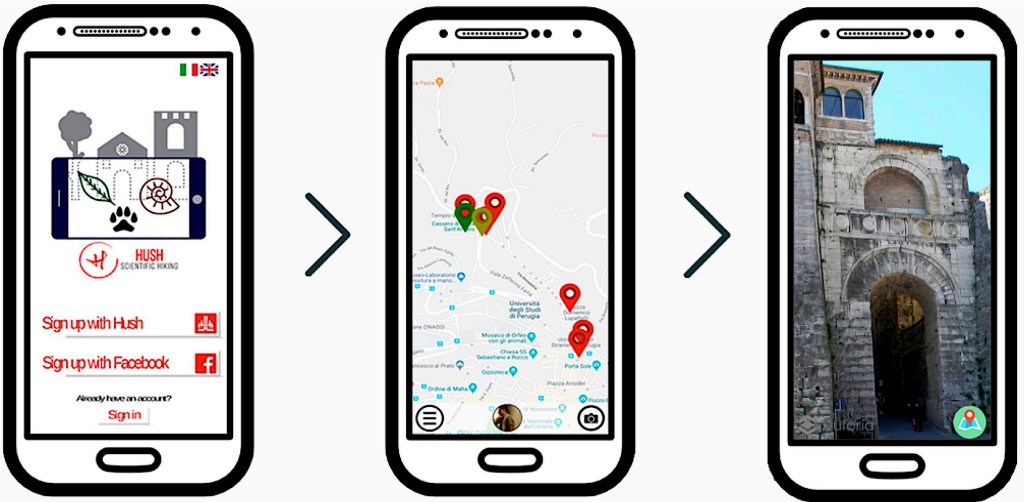

**Figure 3.** The three main steps required for using HUSH. From the left to the right: the login, the research of the PoIs, and the fruition of the multimedia contents.

Initially, the user signs up and logs in. Registration can be accomplished by filling out a questionnaire. Personal information is not required, while some data such as the user's age, sex, and, more important, preferences on a broad spectrum of cultural content are mandatory. These data are used by a machine learning algorithm present in HUSH. The algorithm should work to personalize the experience on the basis of the preferences expressed by the user in the login step. Then, the user is taken to the map section, where the PoIs will be shown. The selection of the PoIs can be conducted according to different options. The first is inclusive of all of them: the map "shows all the PoIs" in the database.

The geological and floristic/faunistic targets are highlighted with different colors. The second option allows the user to "find PoIs around you", and only the PoIs close to the user's position (in a circle with the center corresponding to the visitor's position and a radius of 500 m) are shown. The third option of visualization is for "finding the PoIs on a guided tour". This preference guides the visitor along an already prepared route, suggested according to a keyword-based search. The fourth option allows the user to "search for POIs"; the user can visualize only the PoIs that meet the selected requirements. The values used for the selection are the categories inserted into the database. The fifth and last option answers the command "find PoIs with the intelligent search", where a machine learning algorithm selects only the PoIs matching the preferences of the visitor using a score assigned to each PoI. The rating is given by the multiplication of the percentages of the user preferences with alpha, which is the normalized value of that PoI for the specific label. So, each PoI has a score depending on the user's preferences. The resulting PoIs will be selected and shown on the map.

Once the PoIs are selected, the request is sent to the server, a virtual machine running over the IaaS OpenStack of the Department of Physics and Geology of Perugia and of INFN (Istituto Nazionale di Fisica Nucleare—Perugia Section). The request will be processed by some PHP and Python scripts; in particular, the scripts in Python deal with querying the NoSQL MongoDB database. When a PoI has been selected, the user, with the camera present on their mobile device, targets it (Figure 4).

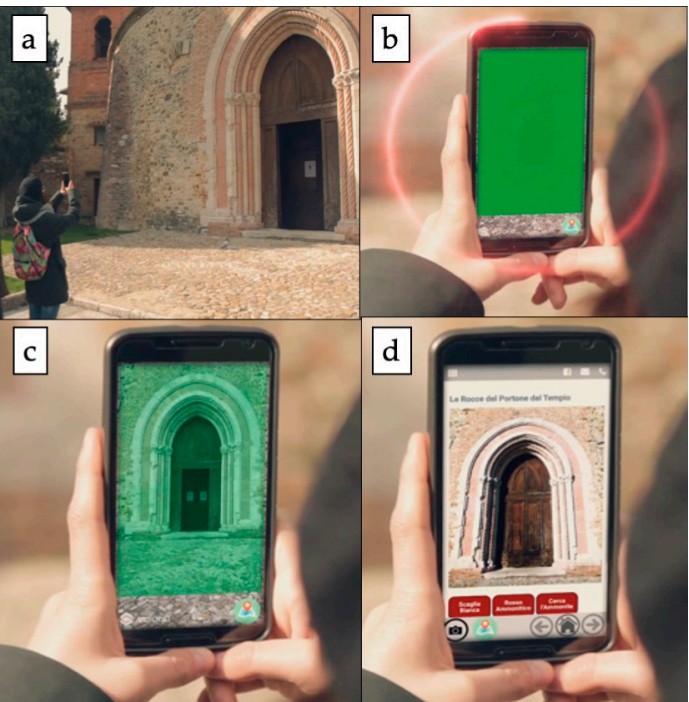

**Figure 4.** The acquisition process of points of interest. (**a**) The visitor finds the PoI; (**b**) the visitor opens the command on the app; (**c**) the app identifies the target image; (**d**) the app opens the contents. The images refer to PoI no. 15 in Tables 1 and 2.

The right framing is suggested in the application for each PoI. One of the characteristics of HUSH is that a visitor can interact with the application, sending feedback on their experience. For this reason, before logging out from the app, a visitor can rate a PoI and suggest new PoIs using a specific functionality named "Scientific reporter". This tool asks for some information about a potential heritage site in the form of a brief and free description. Photos and images can be uploaded too. The last section is devoted to the commercial activities and to the touristic and didactic events in the areas surrounding

the PoIs. This part of the application has the aim of improving and supporting the local economy. Figure 5 shows the full options available when using HUSH.

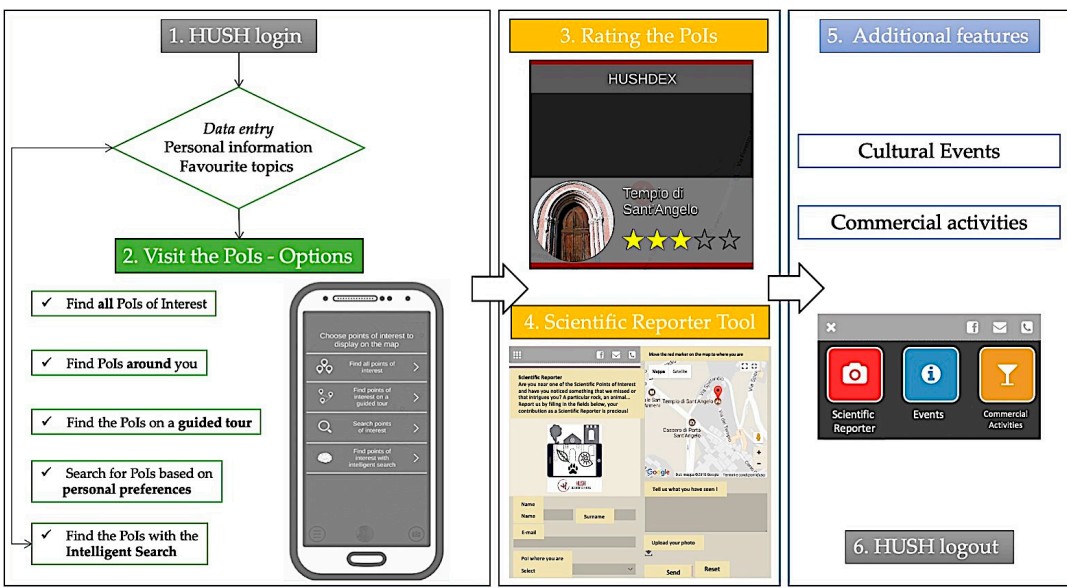

**Figure 5.** The process required for exploring HUSH from point 1 to point 4. (1) HUSH login. (2) Visit the PoIs with the list of different options. All of them require the georeferenced position of the user; in addition, the last option, "*Find the PoIs with the Intelligent Search*", requires personal information given in the data entry step. (3) Rating the PoIs with a score between 0 and 5. (4) The Scientific Reporter window. (5) Additional features such as links to the cultural events and the commercial activities present in the area. (6) HUSH logout.

## 3. Results

Two paths in the downtown area of the city of Perugia have been created. The paths cross the city center in a north to south direction. The first one (path 1 in Table 1) moves from the medieval gate named Porta St. Angelo (43°07′10.4″ N 12°23′04.6″ E) to the Etruscan Arch (43°06′52″ N 12°23′22.8″ E); it is about one kilometer long and it is located in the northern part of the city. Along this path, two biotic PoIs and six geological PoIs are present.

The second path (path 2 in Table 1), 950 m long, moves from the St. Ercolano Church (43°06′31.13″ N 12°23′22.38″ E) to the St. Pietro Church (43°06′05″ N 12°23′44″ E). Along this path, five biotic PoIs and seven geological PoIs are present. Two additional PoIs (one for each path) are introduced to the routes from a historical point of view (see Table 1).

Both the routes follow many historical streets ("royal streets") that have strong meanings in the history of the city. Along these roads, therefore, the historical and architectural heritage is very important and is a common thread and an attraction for the scientific content that HUSH wants to promote.

In Table 2 the different kinds of multimedia content are defined for each PoI.

Below are described some examples of content. In Figure 6 are shown some screenshots captured from one video based on a virtual flight.

A digital elevation model with a 4× exaggeration of the altitude value (a.s.l.) was used to highlight the morphology of the area. The territory toward the north for path 1 is described with the aim of linking the geological characteristics of this area to the social and economic development of this segment of the city throughout the centuries. From the mountain area present toward the north, before the 20th century wood and coal arrived in the city thanks to the abundance of forests. These activities have affected the economic and social development of this part of the city. The visitor can "fly" above the area, without viewing the vegetational and anthropic layers, to better understand the geomorphological parameters. Figure 6 refers to PoI no. 14 in Tables 1 and 2.

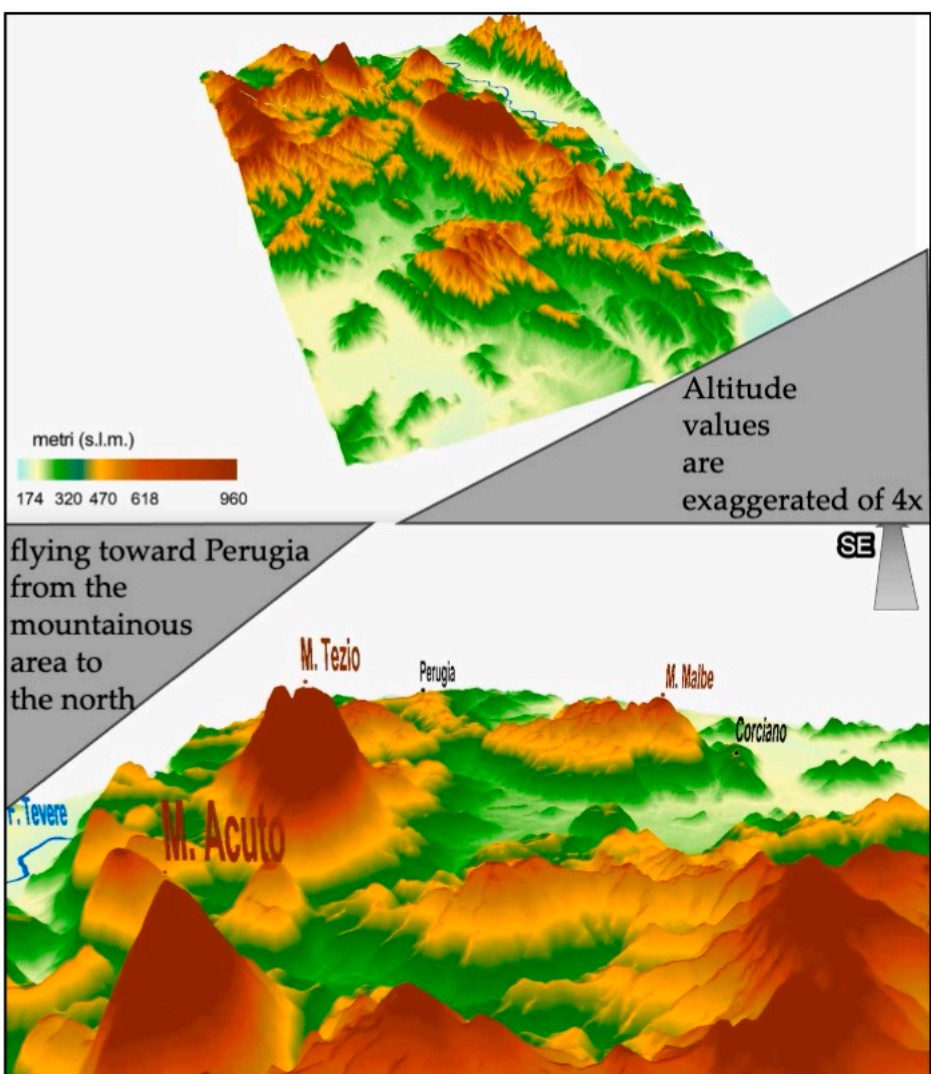

**Figure 6.** A virtual flight that illustrates the morphological arrangement of the area close to the starting point of path number 1. The images refer to PoI no. 14 in Tables 1 and 2.

Figure 7 illustrates an analysis of the topographic arrangement of PoI no. 10 in Tables 1 and 2. The area is a clear example of natural landforms modified by man.

The initial natural landform was a saddle between the threshold area of two streams, both with strong headwater erosion. To avoid erosion and to facilitate the transit of vehicles and people along the main access route to the city from the north, the area was filled. Now, the area can be classified as a natural landform modified by man. The image illustrates the initial DEM created by interpolating the data derived from the geological surveys (Figure 7a), the DEM reproducing the actual topographic surface (Figure 7b), and the area covered by filling material (Figure 7d). The perimeter of the area and the thickness of the deposits are obtained by subtracting the two raster grids. The overlay of some urban buildings (Figure 7c) is necessary for supporting the visitor in orienting himself on the map.

Figure 8 shows an example of augmented reality content, with a 3D model of an ammonite that is present and well-evident on the facade of the S. Angelo Church (PoI no. 15 in Tables 1 and 2).

The visitor is guided in the app first to find the ammonite and then to understand what an ammonite is and why they are present in the rocks of many historical buildings in Perugia and in the Umbria region. It is also possible to open the 3D model to visualize the fossil in a 3D view to better appreciate the morphological details.

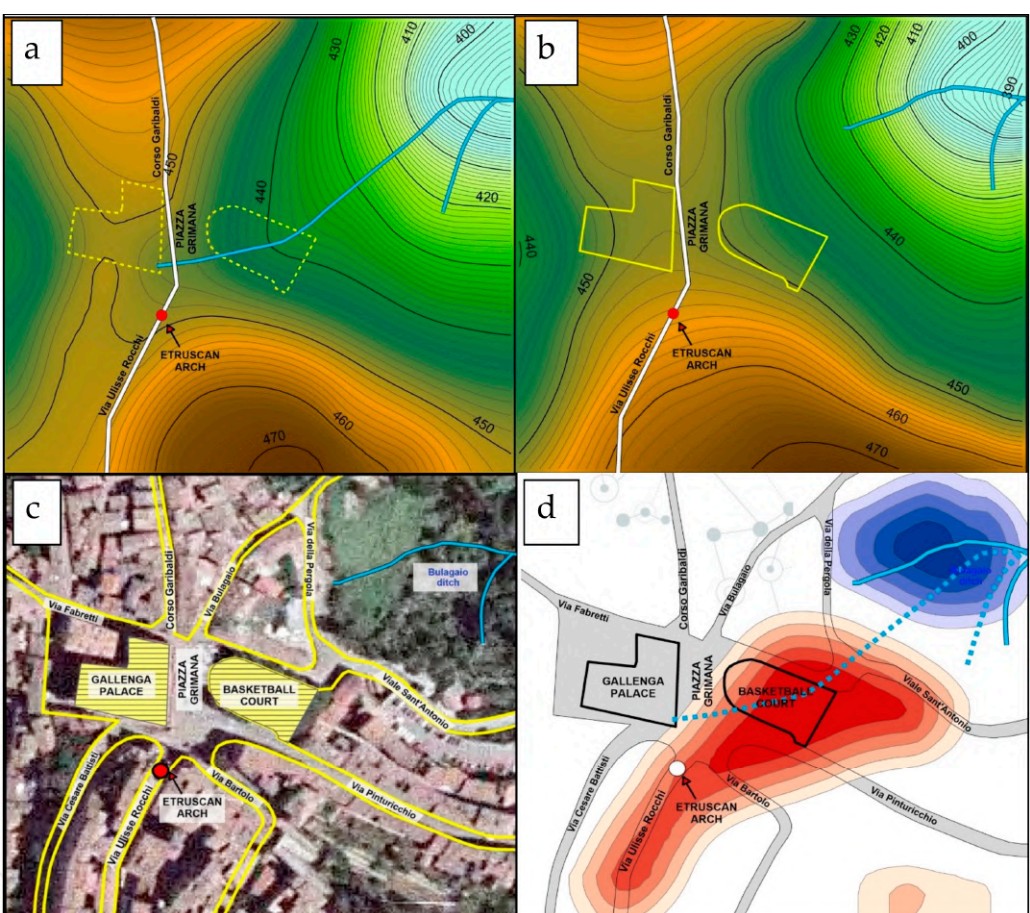

**Figure 7.** The morphological evolution of Grimana Square. (**a**) The figure shows the DEM before the anthropization; (**b**) the DEM after the anthropization; (**c**) an orthophoto of the area; (**d**) the filled areas (graduated scale in red) and the excavated areas (graduated scale in blue). The images refer to PoI no. 10 in Tables 1 and 2.

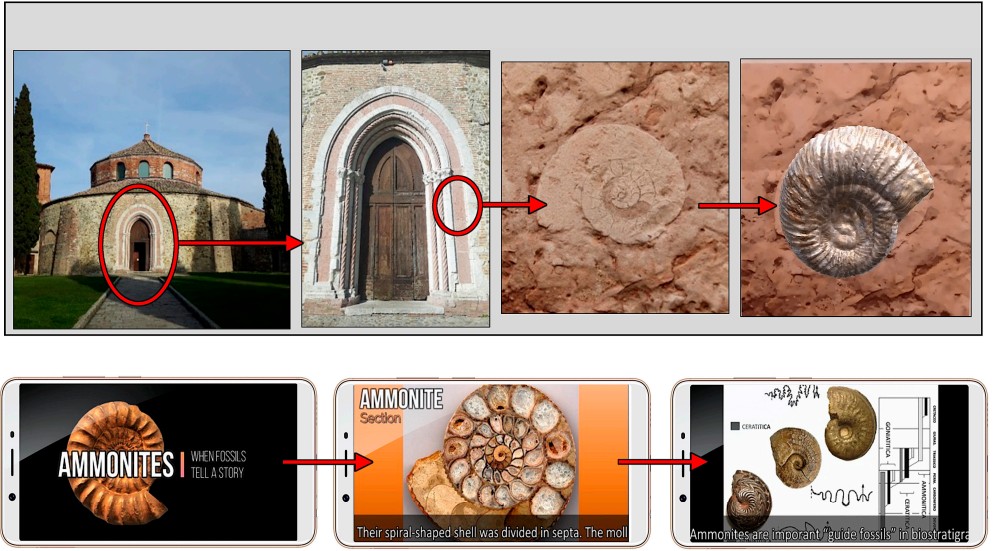

**Figure 8.** The 3D model in augmented reality of a fossil of an ammonite present at the entrance of the St. Arcangelo Church. The images refer to PoI no. 15 in Tables 1 and 2.

## 4. Discussion and Conclusions

The mobile application HUSH has been designed by integrating synergically different professionals using a multi-disciplinary approach that has characterized both the research stage and the application framework. Geologists, naturalists, physicists, and computer scientists have worked together in each step. The topic of the project was to realize an advanced mode of fruition of naturalistic (geological, fauna, and vegetation) and artistic sites in the urban area through the fusion of itineraries of interest and access to augmented reality content (AR).

The naturalistic value of a site is generally associated with sparsely populated areas, while in urban areas different interests prevail. Naturalistic content therefore represents unexpressed potential, which if appropriately valued in AR could represent a strong precursor of development.

With this in mind, the major expected outcomes of HUSH are:

- To identify in an urban context naturalistic values, starting from the historical center and ending in green areas.
- To realize innovative fruition via AR tools of virtually moving in space and time, triggering awareness-raising practices.
- To expand the potential for economic development by linking AR content to local businesses (e-advertising).
- To integrate society into the processes of scientific research.

HUSH guides the user through points of scientific interest in urban areas. The points of interest merge the historical–artistic value of the historic center with its naturalistic values. This is accomplished through innovative technologies such as augmented reality and machine learning techniques. The functioning of the app is very simple. The points of scientific interest are identified and displayed on a map. By clicking on each of them, it is possible to activate a navigation path. The element to be framed is also shown once the PoI is reached, and this allows the user to access scientific content in an augmented reality format. The access to the content is innovative and is directed by framing a physical object, which may be, for example, an architectural element or a landscape. Within the app the information flow is not unidirectional. If the user notices some elements capturing his/her curiosity, he/she can use the functionality of the Scientific Reporter to give us notice, sending us images, text, and position information. In this way, exchange with the users will be encouraged so that the information can grow and be updated also based on the experiences of the users.

HUSH is not limited to urban contexts but can also be used in natural ones too. One of the advantages of this approach is that traditional analog panels and QR codes are avoided. It is well-known that these media deteriorate, can be damaged, and need constant maintenance. Another point in favor of HUSH is that the visitor can be an active player in the process of recognition and in building the contents of the application, not just a simple user. To date, HUSH has already been used in several events involving people of different ages and education levels, with great success. During these testing events HUSH showed only one weakness due to difficulty, reported by a small number of users, in acquiring target images for some PoIs, in particular where well-defined architectonical or artificial elements were missing. In particular this was true close to natural features that changed over time (i.e., trees). This feedback has been taken into account for improving the targets and the database with other images acquired during different time periods. HUSH was used as a tool for urban trekking dedicated to schools of every order and degree, improving the simple oral descriptions. In one of the events involving citizenship, HUSH was used for a treasure hunt where descriptions of PoIs helped in finding the sites needed to achieve the ultimate goal or find a treasure (Figure 9).

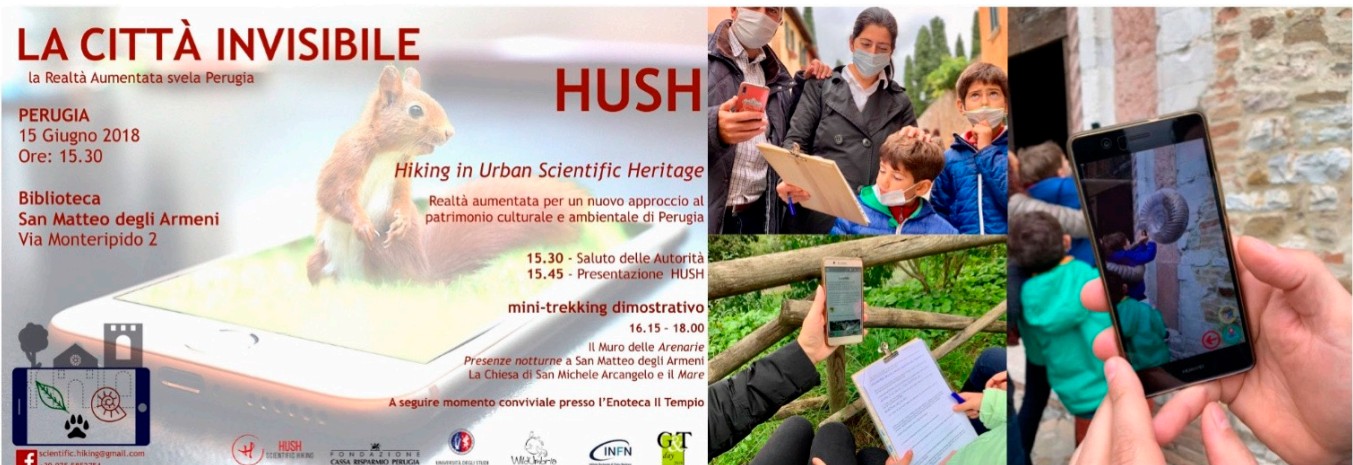

**Figure 9.** One of the outreach events organized using HUSH. The images refer to a treasure hunt along the PoIs in Perugia organized by the POST (Science Museum of Perugia) on 20 October 2020.

The event gathered positive acclaim and fostered interaction between visitors and knowledge acquisition. This approach combined a game and a social network for interpretation and research of geoheritage, confirming itself as one of the most useful approaches to achieving the objectives of dissemination of naturalistic content.

HUSH is free for download for Android (https://play.google.com/store/apps/details?id=com.scientifichiking.HUSH&pli=1, accessed on 1 July 2023) and for Apple (https://apps.apple.com/it/app/hush/id1533334792), accessed on 1 July 2023.

**Author Contributions:** Materials and Methods (the building of PoIs—points of interest), L.M.; and Methods (the creation of the mobile application), G.B. and L.F. The other sections have the same contributions from the authors. All authors have read and agreed to the published version of the manuscript.

**Funding:** This research was funded by the project 'HUSH—Hiking in Urban Scientific Heritage. Realtà aumentata per un nuovo approccio al patrimonio culturale e ambientale urbano di Perugia', 2017.0103.021 Ricerca Scientifica Tecnologica Università degli Studi di Perugia and it is supported by the Fondazione Cassa di Risparmio di Perugia.

**Institutional Review Board Statement:** Not applicable.

**Informed Consent Statement:** Not applicable.

**Data Availability Statement:** Data is unavailable due to privacy restrictions.

**Acknowledgments:** The authors acknowledge the nonprofit organization WildUmbria (http://wildumbria.it/) for the naturalistic content. Accessed on 25 July 2023.

**Conflicts of Interest:** The authors declare no conflict of interest.

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
