# Peer review of "HUSH (Hiking in Urban Scientific Heritage): The Augmented Reality for Enhancing the Geological and Naturalistic Heritage in Urban Areas"

_applsci, doi:10.3390/app13158857_

Round 1

Reviewer 1 Report

Dear Authors,

Thank you for the interesting contribution. Mobile apps play an important role when it comes to education and showcasing the values of the environment. However, the number of such products is still small.

Below are comments that, in my opinion, can improve the quality of the manuscript:

1. Missing from the Introduction is a brief overview of similar applications developed to date. Or more broadly, the use of modern mobile technologies in the presentation of the values of the urban environme

2. It would be advisable to show the general structure of the application in a diagram, perhaps also screenshots of the general application menu.

3. The discussion should point out the strengths and possible weaknesses of the application. Regarding existing products. What sets it apart?

4. Has the app been user tested yet? Have you been able to gather any feedback on its functionality?

5. Unfortunately, the link provided in the manuscript does not work, nor can the app be found in the Play Store. This is a pity because it would allow you to better understand and evaluate the application.

Author Response

Here follow answers to their comments. We denote reviewers’ questions/comments with R and authors’ answers with A.

Reviewing: 1

R1: Missing from the Introduction is a brief overview of similar applications developed to date. Or more broadly, the use of modern mobile technologies in the presentation of the values of the urban environment

A1: In the Introduction a broader overview of the scientific literature on mobile applications, in general and for scientific dissemination in urban areas have been added. Eighteen citations have been added in the references.

R2: It would be advisable to show the general structure of the application in a diagram, perhaps also screenshots of the general application menu.

A2: The new figures 2, 3 and 5 shows the general structure of using HUSH. In figure 5 some screenshots of the app have been added as required from R1.

R3: The discussion should point out the strengths and possible weaknesses of the application. Regarding existing products. What sets it apart?

A3: The strengths of HUSH have been  already listed in the first submitted document from lines 1464 to line 1470. About the weakness: we have tested HUSH in several and different events. We did not have particular problems. Only occasionally some users put in evidence some difficulties in acquiring the target images. This aspect has been inserted in the text from line 1470 to line 1478.

R4: Has the app been user tested yet? Have you been able to gather any feedback on its functionality?

A4: The app has been already tested as highlighted from line 454 to 468 in the first submission. In this second version a last figure (Figure 9) has been added to show some moments of these events.

R5. Unfortunately, the link provided in the manuscript does not work, nor can the app be found in the Play Store. This is a pity because it would allow you to better understand and evaluate the application.

A5: Sorry for the inconvenience ,the links in the first submission are no longer working. New links have been reported from line 1540 to line 1542.

Reviewer 2 Report

My comments and suggestions are presented on the uploaded file

The English needs revision

Author Response

Here follow answers to their comments. We denote reviewers’ questions/comments with R and authors’ answers with A.

Reviewing: 2

R1: The chapter “Materials”, starts by expressing an idea. How can an idea be a material? The authors should try to organize the text in a more logic way, with sub- titles suitable to what is presented. For instance, if they use the term “methodology” instead of dividing the chapter into “materials” and “methods”, would perhaps be more realistic. The term methodology is broader, and they could then make sub-chapters to present the different ideas, contents and parameters, used to build the app.

A1: The first paragraph starts by expressing the idea behind HUSH since it was a sort of introduction to the chapter. The material in our ideas were the contents of the PoIs. However,m in the revised manuscript the chapter is titled “Methodology” as suggested by the Reviewer with the sub-chapters with the same titles as in the first submitted paper but, of course, with a different numbering. 

R2: The English is sometimes hard to follow, it needs a thorough revision.

A2: The English language has been revised.

R3: The terms “geoheritage” and “natural heritage” are used as though they have different meanings, with natural heritage being considered the fauna and flora (living beings). However, geologic aspects are also part of the natural heritage of a site thus, I think the authors should better use the term “natural heritage” to designate the ensemble of geo and bio heritage. And I must say that when I read “Hiking in Urban Scientific Heritage”, it came to my mind that it would be an app to conduct us along museums and scientific fairs; only after reading the article have I understood that by “scientific” the authors meant “natural”, so to me, “Hiking in Urban Natural Heritage”, would suit better the purpose of this app.

A3: In the revised paper a better use of natural and geological heritage has been done changing all the sentences where these terms were present. We thank the Reviewer for the suggestion on the name for HUSH but it could be very difficult now change the name since it is already present online and in some ongoing projects.

R4: If one Point of Interest deals with commercial sites to promote local economy, why not add the location of cultural points of interest too? This would not also promote the economy, because people must pay to visit them, but would also increase people’s knowledge towards the local culture (in which Italy is very rich).

A4: We thank the Reviewer for this suggestion. Our purpose was not to enlarge the contents to the other cultural presences on the territory just to focus our effort on the natural heritage. Nevertheless, the additional tool on commercial activities and events has as main aim to improve the small and local commercial activities with an approach toward the strengthening of local traditions.

R5: “The flow of the app is very simple and it is divided into three main parts:” It would be interesting to present a chart flow describing these aspects. An image is worth a thousand words.

A5: The new figures 2, 3 and 5 shows the general structure of using HUSH. In figure 5 some screenshots of the app have been added as required from R1.

R6: “Two paths in the downtown of the Perugia ci es have been created”; this sentence is presented in the Results, and this other “HUSH has already tested in the urban area of Perugia and two geotouristic routes have been created” appears in the Introduction. So, we either have a spoiler or the results are not quite new.

A6: The Introduction has been modified without anticipating the results.

R7: In the Discussion section, it is said “This project aims to valorize the geo-naturalistic and historical-artistic heritage in urban area”; it seems that the app aims historical and artistic sites too after all, but this is only mentioned now. It is also in this sec on that the objectives underlying the creation of HUSH, are presented. Some of the text presented here is part of the Introduction. Here they should be discussing how this new app surpasses other already in the market, and how it can help to value the natural and historical heritage, of urban areas such as Perugia. Conclude as to the utility of the app for tourists and local inhabitants.

A7: Some sections in the Discussion have been moved to the “Introduction”. In the Discussion the weaknesses of HUSH have been described while the advantages (comparing this app to others already present) have been already put in evidence.

R8: I believe the idea underlying this article is interesting but the way it is presented, is not the best

way to pass the message.

A8: The Authors hope the changes in this second review improve the quality of the paper.

R9: And the English needs a major revision too.

A9: The English has been checked.

Round 2

Reviewer 1 Report

Thank you for the responses to my comments.

Author Response

Dear Reviewer,

Thank you for your support in reviewing the article and for accepting this final version. Some final English mistakes have been erased.

The new submission includes: a revised manuscript with changes tracked in red (applsci-2508118(2)_rev_24-07-2023_with_revision, Word file); a clean version of the revised version (applsci-2508118(2)_rev_24-07-2023_clean) in editable doc file and pdf format too.

Best regards,

Laura Melelli

Reviewer 2 Report

The new version is much better and clearer than the first one, although some minor english mistakes can still be found (e.g. "These information reveals places defines as “morphological false”,...)

As said above, there are still some minor mistakes that need correction

Author Response

Dear Reviewer,

Thank you for your support in reviewing the article and for accepting this final version. Some final English mistakes have been erased as you can see on lines:

22

26

39

48

64

81

130

158

182

201-203

260

274

303

305

324

361

365

367

444

447

466

The new submission includes: a revised manuscript with changes tracked in red (applsci-2508118(2)_rev_24-07-2023_with_revision, Word file); a clean version of the revised version (applsci-2508118(2)_rev_24-07-2023_clean) in editable doc file and pdf format too.

Best regards,

Laura Melelli
